# Selection by a panel of clinicians and family representatives of important early morbidities associated with paediatric cardiac surgery suitable for routine monitoring using the nominal group technique and a robust voting process

Christina Pagel,[1] Katherine L Brown,[2] Isobel McLeod,[3] Helen Jepps,[4] Jo Wray,[2] Linda Chigaru,[2] Andrew McLean,[3] Tom Treasure,[1] Victor Tsang,[2] Martin Utley[1]

► Prepublication history and additional material are available. To view these files please visit the journal online (http://dx.doi.org/ 10.1136/bmjopen-2016-014743).

[1]Clinical Operational Research Unit, UCL, London, UK
[2]Department of Cardiorespiratory, Great Ormond Street Hospital for Children, London, UK
[3]Royal Hospital for Children, Glasgow, UK
[4]Bradford Teaching Hospitals NHS Foundation Trust, Bradford, UK

**Correspondence to**
Dr. Martin Utley;
m.utley@ucl.ac.uk

## ABSTRACT

**Objective** With survival following paediatric cardiac surgery improving, the attention of quality assurance and improvement initiatives is shifting to long-term outcomes and early surgical morbidities. We wanted to involve family representatives and a range of clinicians in selecting the morbidities to be measured in a major UK study.

**Setting** Paediatric cardiac surgery services in the UK.

**Participants** We convened a panel comprising family representatives, paediatricians from referring centres, and surgeons and other clinicians from surgical centres.

**Primary and secondary outcome measures** Using the nominal group technique augmented by a robust voting process to identify group preferences, suggestions for candidate morbidities were elicited, discussed, ranked and then shortlisted. The shortlist was passed to a clinical group that provided a view on the feasibility of monitoring each shortlisted morbidity in routine practice. The panel then met again to select a prioritised list of morbidities for further study, with the list finalised by the clinical group and chief investigators.

**Results** At the first panel meeting, 66 initial suggestions were made, with this reduced to a shortlist of 24 after two rounds of discussion, consolidation and voting. At the second meeting, this shortlist was reduced to 10 candidate morbidities. Two were dropped on grounds of feasibility and replaced by another the panel considered important. The final list of nine morbidities included indicators of organ damage, acute events and feeding problems. Family representatives and clinicians from outside tertiary centres brought some issues to greater prominence than if the panel had consisted solely of tertiary clinicians or study investigators.

**Conclusion** The inclusion of patient and family perspectives in identifying metrics for use in monitoring a specialised clinical service is challenging but feasible and can broaden notions of quality and how to measure it.

## Strengths and limitations of this study

► The nominal group technique, augmented by a robust secret voting process, allowed us to incorporate the perspectives of family representatives and clinicians from different professional groups in selecting early surgical morbidities they felt important to monitor in routine practice.

► The robust voting process used identified group preferences from the preferences expressed by individual panellists and identified where there was a lack of consensus, guiding further discussion.

► One limitation of the approach adopted was that it relied on firm and expert chairing.

► There was some unresolved tension between selecting morbidities clearly attributable to the surgical act and morbidities that are important to families but can be considered to 'come with the territory' of congenital heart disease and its management.

► Relying on a face-to-face approach necessarily limited the size of the panel, and we cannot claim that the priorities and preferences expressed are representative of the respective professional groups and of families in general.

## INTRODUCTION

Early mortality following paediatric cardiac surgery, defined as death within 30 days of surgery or death prior to discharge home, has been the focus of many research studies,[1–10] audit initiatives[6 11–14] and, particularly in the UK, of public scrutiny[1 15–20] over recent decades. However, with early mortality having fallen to 2%–3%,[21] attention has shifted to longer term outcomes and to broadening the assessment of early outcomes to include early morbidities.

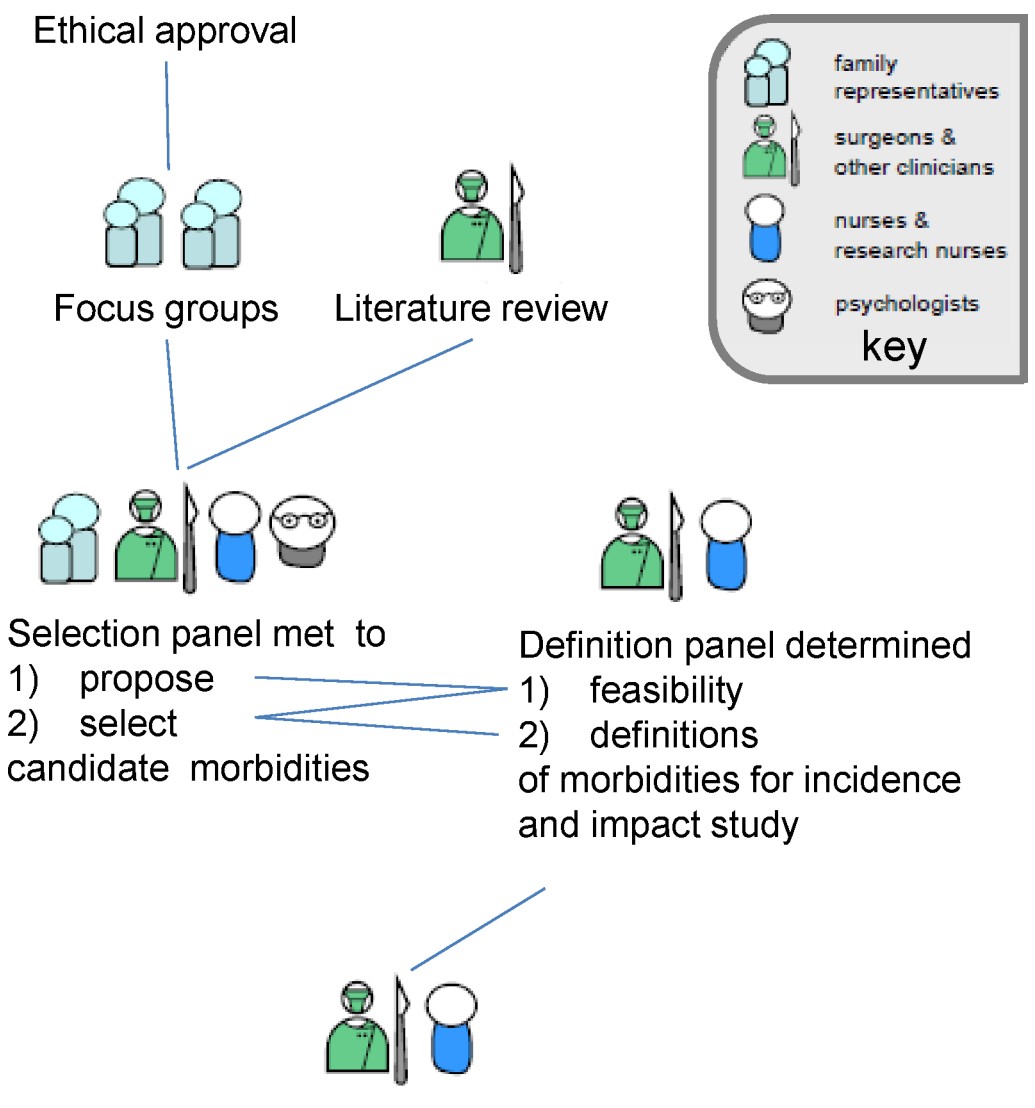

**Figure 1** The role of the selection panel within our wider study to identify and measure the incidence and impact of important morbidities following paediatric cardiac surgery.

Reporting of early morbidities associated with paediatric cardiac surgery has been driven largely by the data available within the databases of professional societies[12] or those available within data sets curated by individual clinical teams.[22] Quality assurance initiatives rooted in such data sets benefit greatly from considerable effort, often over many years, to agree on the definitions of the outcomes collected and design data collection processes. However, it is inevitable that the data sets agreed on, constructed and curated by clinicians (as individuals or via professional societies) focus largely on outcomes considered important from the perspective of that clinician or professional group. Research in other specialties has shown that patients and carers can have quite different perceptions to clinicians on what outcomes are important to monitor as part of service evaluation.[23]

We report here on a process used to select early morbidities as part of a study to identify and then measure the incidence and impact of important early morbidities among paediatric cardiac surgery patients (National Institute of Health Research Health Services and Delivery Research programme (NIHR HS&DR) 12/5005/06).[24] A key aim of our work was to incorporate a broad set of perspectives, including those from family representatives and professionals from different sectors, on what early morbidities were important to monitor in routine practice.

## METHODS
### Overview
Figure 1 gives an overview of the role of our selection process within the wider study. A panel of clinicians and patient representatives met twice to shortlist and then

to select early morbidities, the incidence and impact of which they considered important to measure. As shown in figure 1, our selection process was linked to a parallel process of defining potential morbidities. The shortlist of morbidities produced after the first meeting of the selection panel was considered by a separate group composed entirely of clinicians. This definition panel provided a view on the feasibility of defining, measuring and routinely monitoring each shortlisted morbidity, which informed the second selection meeting.

## Composition of the selection panel

In forming the panel, we aimed to include clinicians from surgical centres, referring hospitals and primary care, as well as family representatives. We wanted a panel of enough people to provide a range of perspectives while being manageable.

The panel comprised 15 people: three family representatives, three paediatric cardiac surgeons, two paediatric intensive care doctors, two paediatric cardiologists, two paediatricians, a paediatric intensive care nurse, a clinical nurse specialist and a clinical psychologist with experience of working with children with congenital heart disease and their families. The panel was chaired by a cardiothoracic surgeon with extensive experience of chairing multidisciplinary panels. Two of the three family representatives were nominated by the Children's Heart Federation, a parent-led charity and umbrella organisation of congenital heart disease charities and voluntary organisations. The third had facilitated one of the focus groups that fed into the selection process. We tried, but did not manage, to recruit a general practitioner to the panel.

With the permission of panel members, both selection panel meetings were recorded and professionally transcribed. Each selection panel meeting also had a predetermined seating plan to ensure that people from similar specialties were not grouped together.

## Selection panel meeting 1: shortlisting

The aim of the first meeting was to identify a shortlist of 15–20 candidate morbidities that would then be considered by the definitions group. Prior to the meeting, the panel was supplied with an extensive list of candidate morbidities identified through the following:

► an ongoing systematic review conducted as part of our wider programme of research;
► three facilitated focus groups held in different UK cities with parents recruited by the Children's Heart Federation; and
► an online forum for patients and families hosted on the website of the Children's Heart Federation.

The focus groups and literature review will be the subject of other publications. The panel was also sent an abridged version of the study protocol, a description of the role of the selection panel and an agenda. Each panellist was asked to identify among or beyond this list of candidate morbidities those they judged most important

to monitor routinely according to a deliberately broad working definition of a surgical morbidity as:

*Any health or emotional problem that arose as a result of the fact of surgery (whether directly caused by surgery/postoperative care or not).*

For the first meeting, panellists were requested not to censor their suggestions on grounds of the perceived difficulty of definition or measurement, and it was stressed that another group would be making these judgements.

We used the nominal group technique[25 26] augmented by a robust voting process to determine group rankings of morbidities. The nominal group technique is designed to reduce the influence of perceived power differentials and of dominant personalities on group decision making while retaining the benefit of discussion absent from other systematic approaches to group decision making such as Delphi.[27 28] We inferred group preferences from individual rankings using a method developed by Utley *et al*,[29] which is briefly described in the online supplementary appendix 1.

## Structure of the first panel meeting

Each panellist was given the opportunity to speak uninterrupted for 2 min on the morbidities they considered important. Each suggestion was entered onto a spreadsheet, which was projected in the meeting room to ensure accurate transcription. The panel was then given the opportunity to add to this initial list if they thought something important had been missed.

The Chair led a process of identifying suggestions that fell outside the working definition above, duplication among suggestions and merging of closely related suggestions. There was then a secret ballot in which panellists were asked individually to rank the resulting list of candidate suggestions in order of descending importance. The voting process and the method for generating group preferences from individual ranking data are described further in the online supplementary appendix 1.

During a scheduled break, the group preferences were calculated from the individual rankings by two of the authors (CP, MU), who did not have a vote on which morbidities to measure. After the break, the group preferences were fed back to the panel. The Chair then led a second round of discussion, focusing on the group preferences and giving panellists the opportunity to argue for specific morbidities being given greater importance, and for the group to further consolidate the list of morbidities.

There was a second round of secret ranking followed by feedback of group preferences prior to a consensus being sought as to the prioritised shortlist of 15–20 morbidities to be passed to the separate definitions panel for an assessment on the feasibility of defining, measuring and monitoring each in routine practice.

## Causal mapping

Following the first selection group meeting, it was decided that it might be useful for non-clinical members

of the panel to have an accessible summary of any causal relationships among candidate morbidities so that, when choosing the best set of morbidities to monitor, any overlap or redundancy among candidates could be accounted for. To this end, a set of causal mapping exercises was conducted by one of the facilitative team (MU) separately with two of the panellists (IM and HJ). In each exercise, cards representing shortlisted morbidities were placed on a large sheet of paper and arranged left to right with lines drawn to indicate potential causal relationships. Photographs were taken of these causal maps, which were then converted to diagrams.

### Selection panel meeting 2: incorporating feasibility and overlap

Prior to the second meeting, panellists were provided with a pack of materials containing:

► a summary of any estimates of incidence and impact of candidate morbidities from the systematic review;
► the judgement of the definition panel on the feasibility of defining, measuring and monitoring routinely each candidate morbidity;
► a summary of potential long-term impacts of each candidate morbidity from the definition panel
► a summary of the parent and family focus groups and the online forum;
► the diagrams generated through the causal mapping exercise;
► minutes from the first meeting;
► a statement of the purpose of the second meeting and its agenda.

At the beginning of the second meeting, the panel was given a brief reminder of the scope of the overall project, the remit of the selection panel and the timescales we were working to.

The panel was tasked with narrowing the list of shortlisted candidates to a selection of 6–10, the incidence and impact of which would be measured in five centres over 18 months.[30] It was explained that the upper limit of 10 morbidities was due to the sample size required for measuring the impact of distinct morbidities. It was explained that we could measure just the incidence of other morbidities if possible from routine data.

Panellists were also alerted to the (then) recently launched NHS England consultation on the future of Children's Heart Services in England, which highlighted the possibility of future national audit of surgical morbidities.

Panellists were asked to consider the following:

► *Feasibility of measurement, including timescales.* Given our plans to measure the impact of morbidities, the project team stressed that selected morbidities needed to be identifiable in a timely manner.
► *Overlap and redundancy among selected morbidities.* The project team made the point that selecting morbidities that almost always occur with other selected morbidities would pose problems in terms of measuring their individual impact. It highlighted

that the length of stay measures are particularly problematic in this respect.
► *Incidence.* Selected morbidities needed to have an incidence of at least 1.5%–2% for us to measure their impact over 18 months due to sample size considerations.

A summary of the judgements of the definition panel was presented to the panel, consisting of an array showing the shortlisted candidates placed vertically based on the group ranking of importance from meeting 1 and horizontally in terms of the feasibility of monitoring that morbidity.

The panel was then asked in a secret ballot to nominate morbidities for exclusion without further discussion and others for inclusion without further discussion. The panel discussed the remaining morbidities as a group.

After the panel meeting, a written summary of the discussion was circulated and an online poll conducted to obtain the group ranking of importance among the shortlisted candidate morbidities. The poll was conducted to elicit the views of panellists who weren't able to attend and to identify replacement morbidities if any of those selected were judged to be infeasible to monitor in routine practice. We used the online voting tool at www.crankit.io, which uses the same algorithm for robustly inferring group preferences from individual preference data that were used in the selection meetings and that are outlined in the online supplementary appendix 1.

### Final review of selected morbidities by definition panel and chief investigators

The selected morbidities were then reviewed by the definition panel, which could veto inclusion of a selected morbidity if, after careful consideration and discussion with the chief investigators (VT, KLB), it was deemed infeasible to define and monitor in routine practice.

### Governance

The study 'Selection, definition and evaluation of important early morbidities associated with paediatric cardiac surgery' received a Favourable Opinion from the National Research Ethics Service London City Road & Hampstead Research Ethics Committee on 8 November 2013 (REC reference 13/LO/1442).

## RESULTS

The panel convened is given in the online supplementary appendix 2. Of the 15 panellists, 3 could not attend the first meeting and 6 could not attend the second. An overview of the results of the selection process is given in figure 2.

### First panel meeting

At the first meeting, 66 morbidity terms were suggested during the round of 2 min contributions from each panellist (see online supplementary appendix 3). In the discussion that followed, seven terms were removed as being irrelevant to our study, related to impacts of

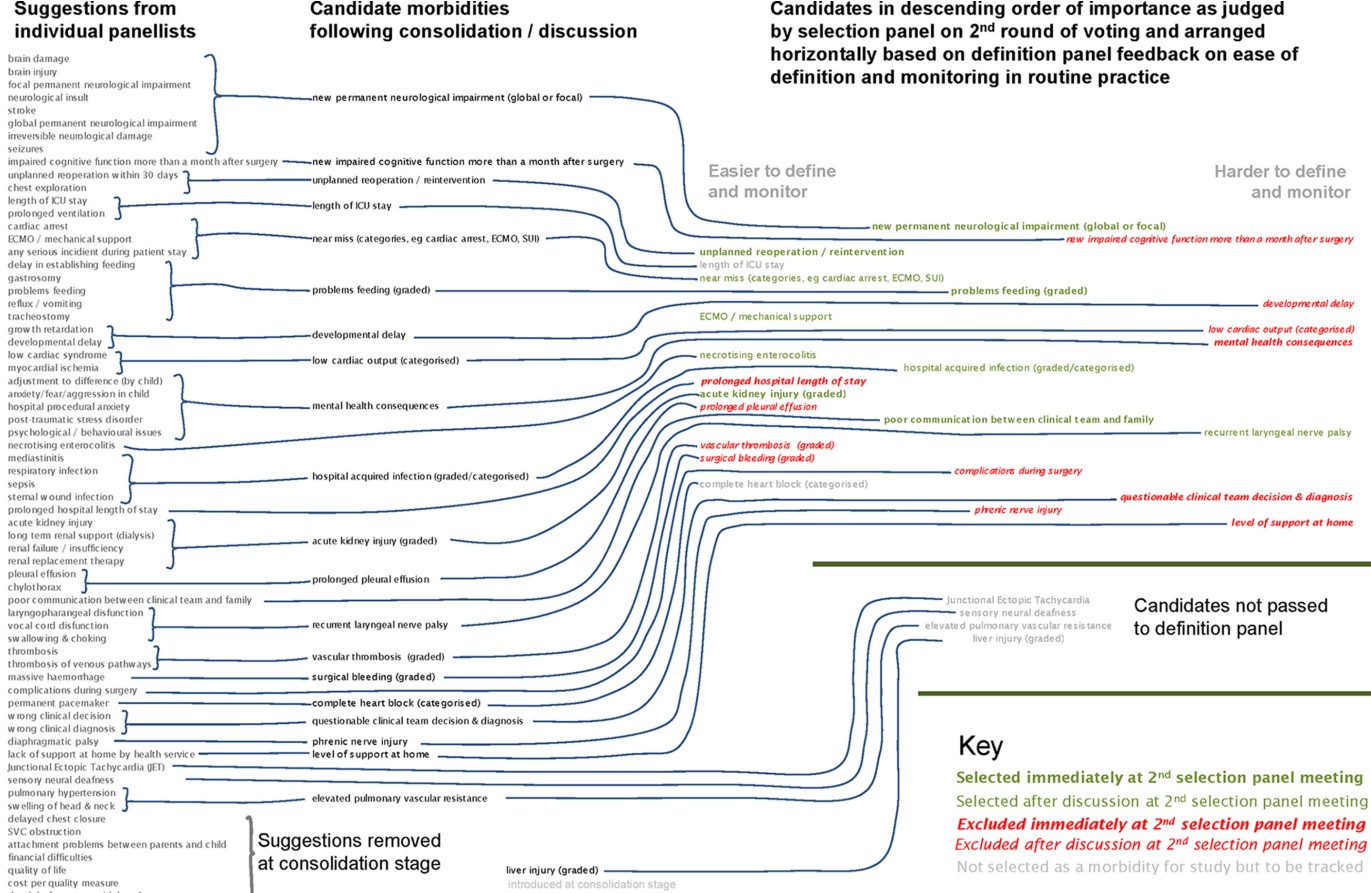

**Figure 2**  Overview of results of selection process.

postsurgical morbidity that would either be measured as part of our empirical study or were too long-term in nature to be captured within our study, or redundant given other terms suggested.

Of the remaining 59, 7 were accepted as candidate morbidities as they were and a further 5 were simply relabelled. The remaining 47 terms were mapped onto 16 groups of 2–8 terms, with each group considered to relate to a sufficiently similar phenomenon for them to be a single candidate morbidity. The members of the panel confirmed that it wanted the term 'ECMO/mechanical support' to feature on its own, as well as being an indicator of a 'major adverse event'. A new term ('liver injury') was added as a candidate at this stage with the agreement of the panel. This gave 29 candidate morbidities.

These candidate morbidities and the group ranking of importance among them in the first round of voting are shown in table 1. It is worth noting that, in this first round of voting, the group ranked 'new global permanent neurological impairment' as the most important morbidity. After that there was a group of 22 candidate morbidities that could only be separated by applying tie-breaks, with the other six candidate morbidities (including 'necrotising enterocolitis') ranked as less important.

In the discussion that followed the first round of voting, the panel merged the two items describing neurological impairment, and individual panellists expressed surprise

at the low ranking of 'necrotising enterocolitis' and 'vascular thrombosis', leading to a discussion of these particular morbidities and their impact.

The results of the second round of voting are shown in table 2. While the panel's view of the top three morbidities (including the merged item of 'new permanent neurological impairment' (global or focal)) was clear, there was a large group of 21 candidates that could only be separated by applying tie-breaks. This group included 'necrotising enterocolitis' and 'vascular thrombosis'. Given the lack of unambiguous group preference among these 21, the panel decided to request that the separate definition panel consider the 24 candidate morbidities given in bold in table 2, with the remainder discarded.

### Second panel meeting

The initial assessment of the definition panel as to the feasibility of defining and monitoring each candidate morbidity still in contention after the first selection panel meeting is shown to the right of figure 2, and as presented to the panel in the online supplementary appendix 4.

Eleven candidate morbidities were considered straightforward to define and monitor in routine practice: 'unplanned reoperation/reintervention', 'length of ICU stay', 'major adverse event (eg, cardiac arrest, ECMO, serious untoward incident)', 'ECMO/mechanical support', 'necrotising enterocolitis', 'prolonged hospital

| Table 1 Results of the first ranking exercise | | |
|---|---|---|
| **Morbidities** | **Rank before tie-breaks** | **Rank after tie-breaks** |
| New global permanent neurological impairment | 1 | 1 |
| New impaired cognitive function more than a month after surgery | 2 | 2 |
| Unplanned reoperation/ reintervention | 2 | 3 |
| Developmental delay | 2 | 4 |
| Major adverse event | 2 | 5 |
| Problems feeding (graded) | 2 | 6 |
| Mental health consequences | 2 | 7 |
| Length of ICU stay | 2 | 8 |
| New focal permanent neurological impairment | 2 | 9 |
| Low cardiac output (categorised) | 2 | 10 |
| Poor communication between clinical team and family | 2 | 11 |
| ECMO/mechanical support | 2 | 12 |
| Acute kidney injury (graded) | 2 | 13 |
| Prolonged pleural effusion | 2 | 14 |
| Complications during surgery | 2 | 15 |
| Hospital acquired infection (graded/categorised) | 2 | 15 |
| Prolonged hospital length of stay | 2 | 17 |
| Complete heart block (categorised) | 2 | 18 |
| Surgical bleeding (graded) | 2 | 19 |
| Questionable clinical team decision & diagnosis | 2 | 20 |
| Level of support from hospital available at home | 2 | 21 |
| Recurrent laryngeal nerve palsy | 2 | 21 |
| Phrenic nerve injury | 2 | 23 |
| Necrotising enterocolitis | 24 | 24 |
| Vascular thrombosis (graded) | 25 | 25 |
| JET | 26 | 26 |
| Elevated pulmonary vascular resistance | 27 | 27 |
| Liver injury (graded) | 27 | 27 |
| Sensory neural deafness | 27 | 27 |

Note that for some suggested morbidities, the panel indicated that a grading or categorisation scheme would be required if that morbidity were to be used. The definition of an appropriate grading or categorisation scheme was left to the separate definitions panel, with only those morbidities selected subject to the full definition process.
ICU, intensive care unit. ECMO, extracorporeal membrane oxygenation, JET, junctional ectopic tachycardia.

length of stay', 'acute kidney injury', 'prolonged pleural effusion', 'vascular thrombosis', 'surgical bleeding' and 'complete heart block'.

| Table 2 Results of the second round of ranking | | |
|---|---|---|
| **Morbidities** | **Rank before tie-breaks** | **Rank after tie-breaks** |
| **New permanent neurological impairment (global or focal)** | 1 | 1 |
| **New impaired cognitive function more than a month after surgery** | 2 | 2 |
| **Unplanned reoperation/ reintervention (categorisation)** | 3 | 3 |
| **Length of ICU stay** | 4 | 4 |
| **Major adverse event** | 4 | 5 |
| **Problems feeding (graded)** | 4 | 6 |
| **Developmental delay** | 4 | 7 |
| **ECMO/mechanical support** | 4 | 8 |
| **Low cardiac output (categorised)** | 4 | 8 |
| **Mental health consequences** | 4 | 10 |
| **Necrotising enterocolitis** | 4 | 11 |
| **Hospital acquired infection (graded/categorised)** | 4 | 12 |
| **Prolonged hospital length of stay** | 4 | 13 |
| **Acute kidney injury (graded)** | 4 | 14 |
| **Prolonged pleural effusion** | 4 | 15 |
| **Poor communication between clinical team and family** | 4 | 16 |
| **Recurrent laryngeal nerve palsy** | 4 | 17 |
| **Vascular thrombosis (graded)** | 4 | 18 |
| **Surgical bleeding (graded)** | 4 | 19 |
| **Complications during surgery** | 4 | 20 |
| **Complete heart block (categorised)** | 4 | 21 |
| **Questionable clinical team decision & diagnosis** | 4 | 22 |
| **Phrenic nerve injury** | 4 | 23 |
| **Level of support at home** | 4 | 24 |
| JET | 25 | 25 |
| Sensory neural deafness | 26 | 26 |
| Elevated pulmonary vascular resistance | 27 | 27 |
| Liver injury (graded) | 27 | 27 |

The morbidities passed to the definition panel for consideration are shown in bold.
ICU, intensive care unit.

Six candidate morbidities were considered less straightforward: 'new permanent neurological impairment (global or focal)', 'problems feeding', 'hospital acquired infection', 'poor communication between clinical team and family', 'complications during surgery' and 'phrenic nerve injury'.

**Box 1    The final list of nine selected morbidities**

► Acute neurological event.
► Unplanned reoperation/reintervention.
► Problems feeding (excluding necrotising enterocolitis).
► Need for renal replacement therapy (excluding extracorporeal membrane oxygenation).
► Major adverse event.
► Extracorporeal life support / extracorporeal membrane oxygenation.
► Necrotising enterocolitis.
► Hospital acquired infection (graded/categorised).
► Prolonged pleural effusion/chylothorax.

The remaining seven candidate morbidities were deemed difficult to define and monitor in routine practice: 'new impaired cognitive function more than a month after surgery', 'developmental delay', 'low cardiac output', 'mental health consequences', 'recurrent laryngeal nerve palsy', 'questionable clinical team decision & diagnosis' and 'level of support at home'.

The output from the causal mapping exercises given in the online supplementary appendix 5 highlighted how 'mental health consequences' could be a result of several other candidate morbidities, how the majority of candidate morbidities could result in prolonged stay in intensive care or hospital, and how 'low cardiac output' could become manifest in several of the other candidate morbidities.

Based on these assessments and the selection panel's own previous assessment of the importance of these morbidities, it was agreed that the following candidate morbidities would be selected without further discussion: 'new permanent neurological impairment (global or focal)', 'unplanned reoperation/reintervention', 'problems feeding', 'acute kidney injury' and 'poor communication between clinical team and family'.

The following candidate morbidities were discarded at this point without further discussion: 'mental health consequences', 'prolonged hospital stay', 'questionable clinical team decision & diagnosis' and 'level of support at home'.

The remaining 15 candidate morbidities were then discussed, with the panel reminded of the need to select at most 5 from these 15. Group decisions were made to select 'ECMO', 'major adverse event', 'necrotising enterocolitis', 'hospital acquired infection' and 'recurrent laryngeal nerve palsy'. Note that the group accepted that 'major adverse event' should not include ECMO as this had been selected as a distinct morbidity.

Group decisions were made to discard at this stage 'new impaired cognitive function more than one month after surgery', 'developmental delay', 'low cardiac output', 'prolonged pleural effusion', 'vascular thrombosis', 'surgical bleeding', 'complications during surgery', 'phrenic nerve injury' and 'length of ICU stay'.

The online poll conducted after the second panel meeting identified 'recurrent laryngeal nerve palsy', 'phrenic nerve injury', 'complete heart block' and 'prolonged pleural effusion' as potential substitute morbidities in the event of the definition panel vetoing any of the selected morbidities in routine practice.

### Final rulings of definitions panel and chief investigators

Of the 10 morbidities selected by the panel, 2 ('poor communication between clinical team and family' and 'recurrent laryngeal nerve palsy') were removed from the final list by the definitions panel following input from and consultation with the chief investigators (VT and KLB) and other members of the project management team. In each case this was done on the grounds that the morbidity concerned would be too problematic to measure in routine practice. The morbidity 'prolonged pleural effusion' was added to the final list of morbidities as a substitute. Feeding problems due to symptomatic recurrent laryngeal nerve palsy were then included under 'problems feeding'. The definition panel and the project team decided to monitor the incidence of phrenic nerve injury and complete heart block by using routinely collected cardiac audit data. These changes were shared with the selection panel.

The final nine morbidities chosen for inclusion in the study, using the revised labels used in working up final definitions, are given in box 1. A detailed description of the definition process and the final definitions used is available in a separate paper.[31]

Alongside these nine, the study team committed to measuring poor communication between the clinical team and the family among the 800+ patients anticipated to enter the matched cohort phase of our study, and to conducting secondary analyses to identify the impact of longer stays in intensive care above and beyond the impact of the selected morbidities.

### DISCUSSION

The morbidities selected to be measured in approximately 3600 patients over 18 months starting October 2015 cover a range of phenomena associated with paediatric cardiac surgery, including indicators of organ damage such as acute kidney injury and necrotising enterocolitis, acute events such as cardiac arrest (included as a 'major adverse event'), the necessity for major interventions in the intraoperative period such as ECMO, and problems feeding that, while not necessarily a priority during the intraoperative period, are considered to have a considerable impact on children and families in the months that follow.

At each stage of selection, the morbidity that was ranked of greatest importance by the panel was neurological impairment. This came as little surprise to the study investigators and vindicates the inclusion in our overarching programme of research of a parallel evaluation of the 'Brief Developmental Assessment', which it is hoped will provide a tool that can be deployed by nursing staff to identify patients who would benefit from referral to specialist neurological or other developmental services.

However, it is fair to say that inclusion on the panel of family representatives and clinicians from outside the tertiary surgical centres brought other issues such as problems feeding and poor communication between clinical teams and families to greater prominence than if the panel had consisted solely of tertiary clinicians or if the study investigators had chosen the morbidities themselves.

We found that opening up the process of choosing the metrics by which services should monitor their performance to include the perspectives of patients and family representatives, which is in line with policy initiatives in England,[32] brought challenges. Throughout our work, there was a tension between choosing a 'clean' set of 'clinical' measures that most closely matched the understanding of 'surgical morbidity' among the tertiary clinicians on the panel and the inclusion of arguably murkier phenomena considered hugely important by families and those working in secondary care.

In particular, those working in surgical centres were more concerned than family representatives and others with the attribution of morbidity to the surgical act, keen to include morbidities that may be related in part to surgical technique (laryngeal nerve palsy and phrenic nerve injury) and degree of success (low cardiac output), and anxious to avoid the attribution to surgical teams of morbidities that are currently considered to 'come with the territory' of congenital heart disease and its surgical treatment. Family representatives and others highlighted the value of gathering information on the incidence and impact of key morbidities, even if they were not caused by surgery, not least as some of them may be reducible through interventions at other points in the care pathway.

## STRENGTHS AND LIMITATIONS

We consider that several features of our study design were key to drawing out and balancing these perspectives. The nominal group technique, starting as it does with an opportunity for each panellist to speak without interruption and within an embedded democratic process, is specifically designed to minimise the influence of perceived power differentials and dominant personalities within a group. This was reinforced by the use of a secret ballot process to determine group preferences, allowing panellists to record their disagreement with the positions stated by others without that being openly declared. The voting tool used distinguishes between unambiguous group preferences and those that rely on tie-breaking. This acceptance and presentation of lack of consensus helped to focus discussion on where it would be most valuable to the task of selecting a group of morbidities and divert unnecessary debate focused on achieving a false consensus through attrition.

Also, our choice to separate the process of identifying which morbidities are most important from the process of assessing the feasibility of defining and monitoring morbidities prevented all parties from self-censoring and

clinical panellists from unconsciously using or claiming privileged knowledge of measurement processes to strengthen the case for morbidities they wanted to include.

Firm and expert chairing was essential to maintaining this discipline. Having a Chair conversant with the clinical area but also experienced in working with multistakeholder groups including patient and family representatives was also key. Although it meant that the Chair brought their own clinical perspective to the table, the panel benefited from the Chair's ability to discern between the wheat and chaff of clinical discussions and summarise for non-clinical participants. It is also questionable whether a Chair that was not an accomplished surgeon and clinical researcher would have held the respect of all parties through the process.

Separating the processes of judging the importance and the feasibility of routinely monitoring morbidities did however risk some of the subtlety of discussions slipping through the gaps between two panels of people. While the preparation of detailed summaries of panel meetings and the presence of the same facilitating team (CP and MU) at all meetings reduced this risk, we acknowledge that the defined morbidities that will be monitored do not correspond exactly in all cases to the phenomena deemed important by the selection panel.

Another limitation of the face-to-face approach adopted is that the panel was necessarily of limited size, and we cannot claim that the priorities and preferences expressed are representative of the respective professional groups and of families in general. For instance, the perspectives of families were represented by three individuals. That said, all three were aware of the concerns of the broader population of families through roles with a parent-led charity or from facilitating one of the focus groups that fed into the selection process.

In summary, we found that the inclusion of patient and family perspectives in identifying metrics for use in monitoring a highly specialised clinical service is a challenging but feasible exercise that can add valuable breadth to notions of quality and how to measure it.

**Acknowledgements** We would like to thank the panellists who contributed to this work, whose details can be found at http://www.gosh.nhs.uk/medical-information/clinical-specialties/cardiothoracic-surgery-information-parents-and-visitors/why-we-do-research/complications-after-heart-surgery-children.

**Contributors** CP, KLB, JW helped design the study, organised and facilitated the panels, contributed to analysis, contributed to the first draft of the paper and approved the final draft. HJ contributed to the causal mapping exercise and approved the final draft. LC reviewed literature for the selection panel and approved the final draft. AM chaired the definition panel that fed into the work and approved the final draft. TT chaired the selection panel, contributed to the drafting of the manuscript and approved the final draft. VT helped design the study as co-chief investigator, contributed to the panel and approved the final draft. MU helped design the study, facilitated the panels, conducted the causal mapping exercise, contributed to analysis, wrote the first draft of the paper and approved the final draft.

**Funding** This project was funded by the National Institute for Health Research Health Services and Delivery Research Programme (Project Number 12/5005/06). MU was in part supported by the National Institute for Health Research (NIHR) Collaboration for Leadership in Applied Health Research and Care (CLAHRC) North Thames at Bart's Health NHS Trust. KLB, JW and VT were supported by the National

Institute for Health Research Biomedical Research Centre at Great Ormond Street Hospital for Children NHS Foundation Trust and University College London. The views and opinions expressed are those of the authors and do not necessarily reflect those of the Health Services and Delivery Research Programme, NIHR, NHS or the Department of Health.

**Competing interests** MU and CP are part of the team that developed the online voting tool CRANKIT used in this study.

**Provenance and peer review** Not commissioned; externally peer reviewed.

**Data sharing statement** There are no additional data available.

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
