## [Reviewer comments · BMJ Open]

ARTICLE DETAILS

TITLE (PROVISIONAL)	Selection by a panel of clinicians and family representatives of important early morbidities associated with paediatric cardiac surgery suitable for routine monitoring using the nominal group technique and a robust voting process.
AUTHORS	Pagel, Christina; Brown, Katherine; Macleod, Isobel; Jepps, Helen; Wray, Jo; Chigaru, Linda; Mclean, Andrew; Treasure, Tom; Tsang, Victor; Utley, Martin

VERSION 1 - REVIEW

REVIEWER	Nikhil Chanani Children's Healthcare of Atlanta Emory University School of Medicine Sibley Heart Center Cardiology 2835 Brandywine Road, Suite 300 Atlanta, Ga 30341
REVIEW RETURNED	16-Nov-2016

GENERAL COMMENTS	Please detail how the participants were chosen. I understand there were three family representatives but I am unclear how they were chosen. This is of concern to make sure it reflects varying levels of socioeconomic status. Individuals of different financial and educational status may value different morbidities.
--

REVIEWER	Lazaros Kochilas Emory University, USA
REVIEW RETURNED	16-Nov-2016

GENERAL COMMENTS	The manuscript is well written and describes a very comprehensive approach to monitoring outcome metrics important to practitioners, families and important stake holders. One item that may need additional clarification and information are the definitions and gradation of the selected morbidities. For example: Prolonged pleural effusion: what is considered prolonged 1 week, > 1 week, > 2 weeks?
---

VERSION 1 – AUTHOR RESPONSE

Reviewer 1

Please detail how the participants were chosen. I understand that there were three family representatives but I am unclear how they were chosen. This is of concern to make sure that it reflects varying levels of socioeconomic status. Individuals of different financial and educational status may value different morbidities.

We agree that this is important detail to include and have added the following sentence to the methods section:

Two of the three family representatives were nominated by the Children's Heart Foundation, a parent-led charity and umbrella organisation of CHD charities and voluntary organisations. The third had facilitated one of the focus groups that fed into the selection process.

To acknowledge that any sample of three family representatives cannot be representative, we have added the following to the discussion of strengths and limitations:

Another limitation of the face-to-face approach adopted is that the panel was necessarily of limited size and we cannot claim the priorities and preferences expressed are representative of the respective professional groups and of families in general. For instance, the perspectives of families were represented by three individuals. That said, all three were aware of the concerns of the broader population of families through roles with a parent-led charity or from facilitating one of the focus groups that fed into the selection process.

Reviewer 2

The manuscript is well written and describes a very comprehensive approach to monitoring outcome metrics important to practitioners, families and important stake holders.

We would like to thank the reviewer for this very positive assessment.

One item that may need additional clarification and information are the definitions and gradation of the selected morbidities. For example:

Prolonged pleural effusion: what is considered prolonged 1 week, > 1 week, >2 weeks?

This detail is available in an open access publication - we have moved the reference to the detailed definitions such that the text immediately preceding table 3 now reads

The final nine morbidities chosen for inclusion in the study, using the revised labels used in working up final definitions are given in table 3. A detailed description of the definition process and the final definitions used is available at [31].

VERSION 2 – REVIEW

REVIEWER	Nikhil Chanani Children's Healthcare of Atlanta Emory University Atlanta, Georgia, USA
REVIEW RETURNED	23-Jan-2017

GENERAL COMMENTS	I think all concerns have been addressed. I wish the parent's were of a broader size but I understand the limitations of the study design.
--

REVIEWER	Lazaros Kochilas Emory University, US
REVIEW RETURNED	27-Jan-2017

GENERAL COMMENTS	I believe the authors should be commented for putting so artfully
---

	together an important quality of care project incorporating care taker and family perspective for such a stressful event as a pediatric cardiac surgery. I still, though, believe that the authors could provide some more information regarding two items:  1. Selection of family members is still not adequately explained how representative is for the group of parents of children with CHD as one of the reviewers requested. This perspective can be different dependent of what for example was the lesion that their child was treated. This may not be easy but some basic background about them would be helpful. For example, the information of which surgical risk categories were belonging their children would be helpful in that regard (were all from the highest surgical risk categories RACHS 5-6 or STAT 5?) 2. Understanding the gradation of the severity of the morbidities even after reviewing the reference 31 is not easy and for some items not readily available. For example how is categorised the complete heart block that recovers within a week of surgery vs the one that requires pacemaker or the low cardiac output. Independent of these minor items that could be addressed with some additional explanations, I believe the manuscript deserves to be published and would be helpful to caretakers and stakeholders related to pediatric cardiac surgery.
--	---

VERSION 2 – AUTHOR RESPONSE

Reviewer 1: Comments to Author:

I think all concerns have been addressed. I wish the parent's were of a broader size but I understand the limitations of the study design.

Reviewer 2: Comments to Author:

I believe the authors should be commended for putting so artfully together an important quality of care project incorporating care taker and family perspective for such a stressful event as a pediatric cardiac surgery.

We would like to thank the reviewer for this positive assessment of our work

I still, though, believe that the authors could provide some more information regarding two items:

1. Selection of family members is still not adequately explained how representative is for the group of parents of children with CHD as one of the reviewers requested. This perspective can be different dependent of what for example was the lesion that their child was treated. This may not be easy but some basic background about them would be helpful. For example, the information of which surgical risk categories were belonging their children would be helpful in that regard (were all from the highest surgical risk categories RACHS 5-6 or STAT 5?)

We did not ask panellists for the RACHS category of the operation that their child had and view this as just one of many potential factors and experiences that might shape perspectives on the importance of different early post-operative morbidities, just as many potential factors and experiences may have shaped the perspectives of each clinician panellist.

Based on the comment of Reviewer 1 above, we feel that we adequately addressed the point he raised at the last iteration about the representativeness of the family members of the panel.

Specifically, the current revision acknowledges that the perspectives of the clinicians and family representatives involved cannot be guaranteed to be representative of their respective wider constituencies whilst giving detail on how the family representatives were recruited to the panel and their credentials for being aware of the perspectives of the broader population of families and carers.

The relevant text is reproduced below

Two of the three family representatives were nominated by the Children’s Heart Foundation, a parent-led charity and umbrella organisation of CHD charities and voluntary organisations. The third had facilitated one of the focus groups that fed into the selection process. (Methods section)

Another limitation of the face-to-face approach adopted is that the panel was necessarily of limited size and we cannot claim the priorities and preferences expressed are representative of the respective professional groups and of families in general. For instance, the perspectives of families were represented by three individuals. That said, all three were aware of the concerns of the broader population of families through roles with a parent-led charity or from facilitating one of the focus groups that fed into the selection process. (Strengths and limitations section of Discussion)

2. Understanding the gradation of the severity of the morbidities even after reviewing the reference 31 is not easy and for some items not readily available. For example how is categorised the complete heart block that recovers within a week of surgery vs the one that requires pacemaker or the low cardiac output.

We apologise for the residual confusion here, which we think is down to us not making explicit what was meant by attaching the qualifier “graded” or “categorised” to morbidities suggested in the first stage of the process. We have added the following text to the caption of Table 1.

Note that for some suggested morbidities, the panel indicated that a grading or categorisation scheme would be required if that morbidity were to be used. The definition of an appropriate grading or categorisation scheme was left to the separate definitions panel, with only those morbidities selected subject to the full definition process.

Independent of these minor items that could be addressed with some additional explanations, I believe the manuscript deserves to be published and would be helpful to caretakers and stakeholders related to pediatric cardiac surgery.

VERSION 3 – REVIEW

REVIEWER	Lazaros Kochilas Emory University, Atlanta, USA
REVIEW RETURNED	06-Mar-2017

GENERAL COMMENTS	I believe the authors addressed all points as far as possible within the scope of their work and their manuscript will be of interest to the readers of BMJ Open.
---